# A Three-Dimensional Buffer Analysis Method Based on the 3D Discrete Global Grid System

Jinxin Wang [1], Yan Shi [2,*], Zilong Qin [2], Yihang Chen [2] and Zening Cao [2]

[1] School of Earth Science and Technology, Zhengzhou University, Zhengzhou 450001, China; jxwang@zzu.edu.cn
[2] School of Water Conservancy Engineering, Zhengzhou University, Zhengzhou 450001, China; qinzilong@gs.zzu.edu.cn (Z.Q.); cyh2018@gs.zzu.edu.cn (Y.C.); zjczn@gs.zzu.edu.cn (Z.C.)
* Correspondence: shiyan2019@gs.zzu.edu.cn; Tel.: +86-131-2508-0477

**Abstract:** Three-dimensional (3D) buffer analysis is among the basic functions of 3D spatial analysis, and it plays an important role in 3D geographic information systems. The rapid development of the 3D Discrete Global Grid System (DGGS) provides a new method for the 3D spatial analysis of geographic information. According to the spatial topology characteristics of the 3D DGGS and the concept of dimensionality reduction, a 3D buffer analysis method based on the spatial grid of the Earth system is proposed to solve the problem of the buffer algorithm of a space object being unsatisfactory at present. In this paper, we present a method to calculate the distance between cells based on the side length of the spherical grids according to the geometric characteristics of the grids. For the grids of a geographic object, we describe the Euclidean distance expansion algorithm and the radial elevation expansion algorithm that helped us to obtain its 3D buffer. Finally, in terms of algorithm complexity and visualization effect, compared with the traditional raster buffer algorithm, the method presented in this paper has lower complexity, an improved visualization effect, and stronger generality.

**Keywords:** 3D discrete global grid system; 3D modeling; buffer analysis; dimensionality reduction; Euclidean distance

## 1. Introduction

Spatial analysis is among the core functions of geographic information systems, and it is also important for evaluating the function of geographic information systems. Buffer analysis is an important part of spatial analysis. It describes the influence of geo-spatial objects on the surroundings, and it is an effective means to solve the problem of spatial proximity [1]. From a mathematical point of view, the basic concept of buffer analysis is to determine the neighborhood of a given buffer radius and a spatial object or collection. As the expression results of buffer analysis are intuitive and conducive to monitoring, buffer analysis plays an important role in mineral resource evaluation, road traffic management, urban planning, environmental monitoring, and many other fields [2–5].

Buffer analysis mainly includes point, line, plane, and volume buffers according to the type of spatial objects. For two-dimensional space, the buffer generation algorithms and analytic applications are very mature. Buffer generation algorithms are mainly divided into raster algorithms, such as the mathematical morphology expansion method, filling algorithm, and raster-based Euclidian distance transformation, and vector algorithms, such as the parallel double-line method and boundary tracking method [6]. Sumeet et al. studied the construction method of composite buffers for multiple vector objects in vector layers [7]. For special objects, such as polygons with holes, Mate et al. comprehensively considered the analysis efficiency and the practical properties of the analysis object, and they listed different analysis methods [8]. Yang et al. proposed an algorithm for the fast implementation of the buffer analysis of vector data using a Graphics Processing Unit

(GPU), which solved the problem of data self-intersection in the buffer analysis of the traditional vector algorithm [9]. The principle of the raster algorithms is relatively simple and easy to implement, but they have disadvantages, such as low precision and being limited by computer memory. Vector algorithms are usually more complex, involving the calculation and judgment of multiple spatial relations, such as curve intersection, arc cutting and recombination, and inclusion relation determination. The implementation of the algorithms is more complex, but they have the advantage of not reducing the accuracy of original data within the range of computer accuracy, and the memory consumption is much lower than that for the raster algorithms.

With the rapid development of 3D GIS and the wide application of 3D visualization and analysis technology in various fields, 2D buffer analysis has been unable to meet its needs, and 3D buffer analysis has become an urgent requirement. The geometric relations of objects in 3D space are far more complex than those in 2D space, and an algorithm for generating a 2D buffer cannot be directly applied to objects in 3D space. There have been few studies on algorithms for generating a 3D buffer at present. Li et al. used the raster model to express 3D buffers and proposed a raster-based isosurface expansion 3D buffer generation algorithm [10]. The efficiency of the raster-based algorithm is related to the buffer radius. The larger the buffer radius, the greater the number of cells in the grid, the larger the computer memory occupation, and the lower the efficiency. In order to solve this problem, Yuan et al. proposed the introduction of parallel computing into buffer computing, the core concept of which is to decompose the computing part with parallel characteristics into multiple tasks for parallel computing, so as to improve its operating efficiency [11]. Zhang et al. proposed a 0–1 interchange algorithm based on linear octree neighborhood analysis to construct a 3D buffer of urban rail transit [12]. Li proposed a 3D buffer analysis method based on the concept of space filling. After the surface model was discretized, the voxels were screened based on the 3D signed Euclidean distance algorithm, and finally, the 3D buffer surface was reconstructed [13].

The Discrete Global Grid System (DGGS) is a kind of geosphere fitting grid based on the sphere that can be infinitely subdivided but does not change its shape. It has the characteristics of hierarchy and global continuity. When subdivided to a certain extent, it can achieve the purpose of simulating the Earth's surface. It is expected to fundamentally solve the problems of data fracture, deformation, and topological inconsistency of the planar grid model in global multiscale data management [14]. In recent years, international academic circles have conducted in-depth research on the DGGS from different aspects. Effective progress has been made in the hierarchical subdivision and indexing of global spatial data [15,16], global meteorological and hydrological simulations [17,18], spatial data quality and cartographic synthesis [19], global mass image management [20,21], and global visualization models [22,23]. Bowater realized the visualization and query of the offset region around IoT points based on the Rhealpix DGGS [24]. With the development of Digital Earth research, the concept of the Earth System Spatial Grid (ESSG) [25] has been proposed, which is now also called the 3D DGGS. The 3D DGGS belongs to a 3D volume grid, which provides a new method for the 3D visual modeling and analysis of 3D space objects. Chinese scholars have proposed the Spheroid Degenerated-Octree Grid (SDOG) [25], Sphere Shell Space Grid (S3G) [26], Sphere Geodesic Octree Grid (SGOG) [27], and other schemes. Multiscale 3D modeling and visualization of the global lithosphere [28], 3D modeling and analysis of geological truth [29], integrated modeling of the crustal atmosphere and dynamic visualization of elements [30], space orbit target management and collision detection methods [31], the Earth system framework and its application to cellular automata [32], 3D grid technology and its coding conversion in the geospace and lunar sphere [33], etc., have been applied and are in leading positions in the field globally.

With the growing interest in Digital Earth and the subsequent use of the 3D DGGS to meet the needs of efficient indexing, analysis, and visualization of 3D data, in this paper, we propose a new method to generate a 3D buffer to analyze the proximity of spatial objects, for which we chose the SGOG as the data model. The SGOG's sphere division

based on the QTM makes full use of the characteristics of the great circle. Its grid system has the characteristics of simplicity, regularity, and moderate deformation, which can be used for the organization and management of space data integration of space and the Earth. First, the target object is converted to the SGOG. Then, using the spherical topological properties of the SGOG and the radius of the buffer, we propose an expansion method base on QTM edge neighbors to obtain the spherical 2D buffer characteristic grid. Finally, the spherical 2D buffer data are expanded by the method of radial elevation expansion to obtain the elevation data of the corresponding cell IDs, namely, the SGOG of the 3D buffer, so as to generate the 3D buffer of the target object. In order to verify the effectiveness of this method, we compare the proposed algorithm with the raster model buffer in terms of visualization effect and algorithm complexity. The methods proposed in this paper facilitate new ways to obtain 3D buffers to exploit 3D spatial data that are stored in the 3D DGGS framework of a Digital Earth, which is beneficial to spatial analysis and calculation of the 3D DGGS.

The rest of this paper is organized as follows: In Section 2, we describe the data model of SGOG and propose the methods of generating the 3D buffer. In Section 3, we present a variety of results produced by applying the proposed algorithm to obtain the 3D buffer. In Section 4, we conclude the paper and provide directions for future work.

## 2. Methods

A 3D buffer is a 3D object established by a growth element through 3D space analysis, in which a growth element can be a single point, line, plane, or body, or a collection of them. Its mathematical definition is as follows: the 3D buffer D based on the growth element A is $D = \{x | d_{min}(x, A) \leq r\}$; that is, the 3D buffer of radius r for A is the set of all x with minimum Euclidean distance $d_{min} \leq r$ from A.

A Digital Elevation Model (DEM) was used as the test target for our 3D buffer operations. The DEM realizes a digital simulation of the ground terrain through limited terrain elevation data, which is a digital expression of the terrain surface morphology. First, the target DEM data were processed and converted into SGOG cell IDs and the corresponding elevation storage of the cell IDs. If the corresponding target object was a 2D surface object or 3D volume object, it was necessary to extract the boundary data of the target object, convert it into the SGOG, and store the data of its boundary and target region separately. Then, the transformed SGOG cell IDs were transformed by our proposed Euclidean distance expansion algorithm based on QTM cell edge neighbors, and the corresponding cell IDs and the Euclidean distance relationships of the corresponding target cells were obtained, i.e., the 2D spherical buffer of the target object. Then, our proposed radial elevation transfer expansion algorithm was used to expand the corresponding radial elevation of the 2D spherical buffer grids, and the grids' cell IDs in SGOG and corresponding elevations of the 3D buffer were obtained. Finally, the obtained boundary buffer and the radial buffer data set of the target area were intersected to obtain the 3D buffer of the target object.

### 2.1. Data Model Selection and Data Processing

In this paper, the sphere grid of SGOG was selected as the data storage and expression model. The model first splits the recursive quaternary of a spherical large-arc quaternary triangular mesh and then splits the radial equidistant recursive binary [29]. The structures of the grid elements are spherical triangular pyramids around the center of the sphere and spherical triangular platforms at the remaining positions. In view of the fact that the current research content does not reach the geocentric domain, the SGOG units involved in this paper were all spherical triangular prism structures.

Since there are many calculations for the elevation of the radial direction in the calculation of the 3D buffer, if the data are stored in the form of SGOG 'cell ID + radial ID', a large number of radial ID decoding operations will be generated. In order to reduce unnecessary radial ID decoding operations and improve computational efficiency, the data storage format adopted in this paper was 'cell ID + elevation'. Through the expansion

algorithm, all the cell IDs of the object's 3D buffer and the corresponding upper and lower elevations were obtained and finally returned to the SGOG model for visualization.

### 2.2. The 3D Buffer Generation Algorithm Based on QTM Edge Neighbors

Buffer generation can be regarded as starting from the initial cell and gradually expanding to the surrounding neighborhood until the distance from the expanded cell to the initial cell is greater than or equal to the given buffer radius. According to the radial continuity of the SGOG, in this paper we use the concept of dimensionality reduction to transform the 3D expansion of the 3D buffer into the 2D expansion of the growth grid element and the corresponding radial expansion of the elevation. On the one hand, we propose a method of Euclidean distance expansion based on QTM cell edge neighbors to expand the initial cell in 2D. On the other hand, we propose a method of radial elevation transfer expansion to determine the radial elevation of the corresponding 3D buffer cell ID.

#### 2.2.1. The Euclidean Distance Expansion Based on QTM Cell Edge Neighbors

The purpose of the Euclidean distance expansion is to expand the initial cell by using QTM cell edge neighbors to obtain cells with different proximity and their corresponding Euclidean distances, that is, to obtain the spherical 2D buffer zone of the initial cell and the corresponding Euclidean distance attributes. In this paper, the distance between the center points of the QTM cells was used to express the Euclidean distance between the cells.

In order to facilitate the calculation of the Euclidean distance of the grid, complex units were used to indicate the expansion direction of the edge proximity of the grid, and the Euclidean distance between the cell and the initial cell was determined while the edge proximity expansion was performed. In view of the difference between the edge proximity of the positive and negative grids, a complex plane was established, as shown in Figure 1 below. The three complex units of the positive grids are $v_{11}$, $v_{12}$, and $v_{13}$, and the three complex units of the negative grids are $v_{21}$, $v_{22}$, and $v_{23}$. Their values are shown in the following Equation (1):

$$\begin{cases} v_{11} = \cos\frac{\pi}{6} + i\cdot\sin\frac{\pi}{6}, \ v_{12} = -i, \ v_{13} = -\cos\frac{\pi}{6} + i\cdot\sin\frac{\pi}{6} \\ v_{21} = -v_{12}, \ \ v_{22} = -v_{13}, \ \ v_{23} = -v_{11} \end{cases} \tag{1}$$

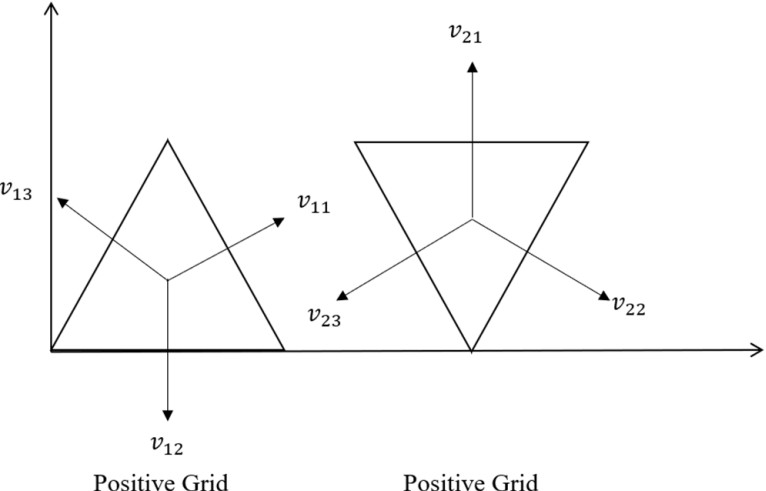

**Figure 1.** The complex units of the positive and negative grids.

First, it is determined whether the initial cell is a positive grid or a negative grid, so as to determine the complex units when the cell expands by edge neighbors. If it is a positive grid, the initial complex units are $v_{11}$, $v_{12}$, and $v_{13}$; if it is a negative grid, the initial complex units are $v_{21}$, $v_{22}$, and $v_{23}$. Then, according to the initial complex unit of the grid, the expansion complex unit of the cell is determined according to the corresponding

relationship in Table 1, and the next step is to expand the edge proximity. Then, according to the initial complex unit of the cell and the corresponding relationship in Table 1, the expansion complex unit of the cell is determined for the next edge to expand using the edge neighbor. Assuming that the steps of complex units $v_{11}$, $v_{12}$, $v_{13}$, $v_{21}$, $v_{22}$, and $v_{23}$ passed by the cell are $k_0$, $k_1$, $k_2$, $k_3$, $k_4$, and $k_5$, the path complex number corresponding to the cell is shown in Equation (2).

$$v = k_0 v_{11} + k_1 v_{12} + k_2 v_{13} + k_3 v_{21} + k_4 v_{22} + k_5 v_{23} \tag{2}$$

**Table 1.** The relationship between the expansion complex unit and the initial complex unit.

| Initial Complex Unit | Expansion Complex Unit |
|:---:|:---:|
| $v_{11}$ | $v_{21}$, $v_{22}$ |
| $v_{12}$ | $v_{22}$, $v_{23}$ |
| $v_{13}$ | $v_{21}$, $v_{23}$ |
| $v_{21}$ | $v_{11}$, $v_{13}$ |
| $v_{22}$ | $v_{11}$, $v_{12}$ |
| $v_{23}$ | $v_{12}$, $v_{13}$ |

We combine Equations (1) and (2) to obtain Equation (3).

$$v = (k_0 - k_5)v_{11} + (k_1 - k_3)v_{12} + (k_2 - k_4)v_{13} \tag{3}$$

From Equation (3), it can be seen that the simultaneous appearance of $v_{11}$ and $v_{23}$, $v_{12}$ and $v_{21}$, and $v_{13}$ and $v_{22}$ would result in a large number of repeated cells during edge neighbor expansion. Therefore, the following provisions are made during expansion: when the path complex number appears to be $v_{11}$, delete $v_{23}$ from the candidate expansion complex units; when the path complex number appears to be $v_{23}$, delete $v_{11}$ from the candidate expansion complex units; the same rules apply to $v_{12}$ and $v_{21}$, as well as $v_{13}$ and $v_{22}$. In this regard, each path complex number is only composed of the sum of three complex units: $v_{11}$ or $v_{23}$, $v_{12}$ or $v_{21}$, and $v_{13}$ or $v_{22}$. The path complex number composition and distribution of the expanded cell are shown in Figure 2. Therefore, $x = k_0 - k_5$, $y = k_1 - k_3$, and $z = k_2 - k_4$ can be set; then Equation (3) can be transformed into Equation (4).

$$v = xv_{11} + yv_{12} + zv_{13} \tag{4}$$

In Equation (4), when $x \geq 0$, this indicates that the complex unit of the number of steps x of the cell is $v_{11}$; when $x < 0$, this indicates that the complex unit of the number of cell steps $|x|$ is $v_{23}$. The relationship between the values of y and z and the complex unit is the same. The expansion complex units and corresponding (x, y, z) value changes are shown in Table 2. At this time, the (x, y, z) value of the initial cell is (0, 0, 0); then (x, y, z) can be used to represent the cell obtained by the path complex number, and they are in a one-to-one correspondence. Incorporating the value of each complex unit into Equation (1), the relationship between the path complex number amplitude and (x, y, z) is shown in Equation (5).

$$|v|^2 = (x - z)^2 \cos^2 \frac{\pi}{6} + \left(y - (x + z)\sin \frac{\pi}{6}\right)^2 \tag{5}$$

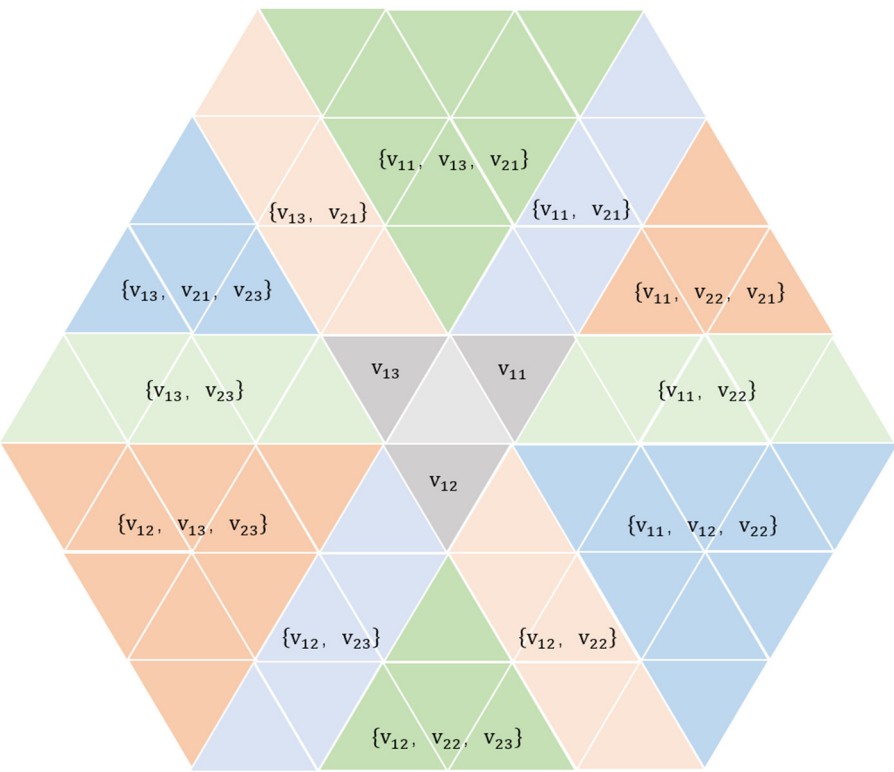

**Figure 2.** The grid's expansion complex unit distribution map.

**Table 2.** Correspondence between the (x, y, z) value and the expansion complex unit.

| Expansion Complex Unit | (x, y, z) |
|:---:|:---:|
| $v_{11}$ | x = x + 1 |
| $v_{12}$ | y = y + 1 |
| $v_{13}$ | z = z + 1 |
| $v_{21}$ | y = y − 1 |
| $v_{22}$ | z = z − 1 |
| $v_{23}$ | x = x − 1 |

If the side length of the grid under this subdivision level is r, the relationship between the distance $l$ between the cell corresponding to the complex number of paths and the initial cell and the path amplitude is as follows: $l = |v|*r/\sqrt{3}$. We can combine this with Equation (5) to obtain Equation (6).

$$l = r\sqrt{\frac{3(x - z)^2 + (2y - (x + z))^2}{12}} \tag{6}$$

As shown in Figure 3, the edge neighbor expansion starts from the initial cell with the cell ID of 120, which is a positive grid. The figure shows the edge neighbor expansion along the $v_{21}$ direction as an example. Corresponding to Table 1, the next expansion complex units are $v_{11}$ and $v_{13}$. According to this recurrence, the path complex number of the corresponding cell and the Euclidean distance between the cells are obtained. This method directly obtains the Euclidean distance between the cells through the cell radius of the corresponding subdivision level and the path complex number between the corresponding cells, avoiding a large number of calculations caused by cell decoding to obtain coordinate points. In the figure, the path from cell ID 120 to cell ID 113 is the complex unit sequence $v_{21}$, $v_{11}$, $v_{21}$, $v_{11}$, and $v_{22}$. When cell ID 120 passes through the complex units $v_{21}$, $v_{11}$, $v_{22}$, and $v_{11}$ to obtain cell ID 103, the next complex unit is $v_{22}$ to obtain cell ID 131 because cell ID 113 has already been obtained. At the same time, the (x, y, z) value of cell ID 113 can be

obtained as $(2, -2, -1)$; by substituting this into Equation (6), the distance between cells 113 and 120 is $l = r\sqrt{13/3}$.

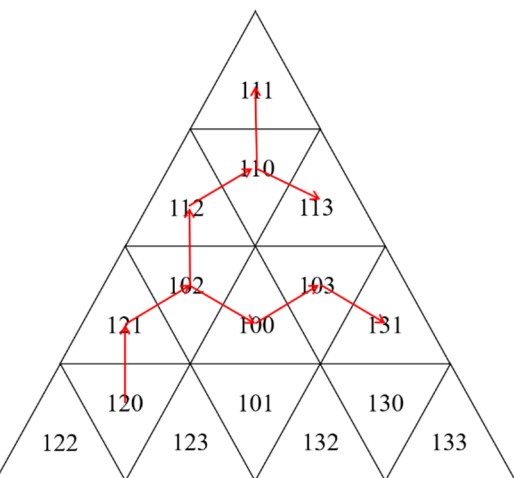

**Figure 3.** Edge neighbor expansion in the grid.

In order to facilitate the storage of the cell ID and the distance data of the corresponding initial cell, cells at the same distance from the initial cell are stored in the same container. To ensure the convenience of storage of the distance data in the container, marking $r = 1$, the distance in the container is represented by $d = 3l^2 = |v|^2$. As shown in Figure 3, assuming that the initial cell's cell ID is 100, the cells with cell IDs 101, 102, and 103 are stored in a container with a distance of 1, and the cells with cell IDs 112, 113, 121, 123, 131, and 132 are stored in a container with a distance of 3, etc. This data storage method facilitates the subsequent radial elevation expansion processing of the cells that are at the same distance from the initial cell.

### 2.2.2. The Radial Elevation Expansion

The function of the radial elevation transfer expansion of the grid is to expand the elevation of the QTM 2D buffer grid obtained by the edge neighbor expansion combined with the buffer radius and transform it into a SGOG 3D voxel. According to the edge neighbor expansion, the distance between the cell and the initial cell is 2D, and the obtained cell's ID is the projection of the 3D buffer area of the initial cell in the spherical direction. It is assumed that the corresponding upper and lower elevations of the initial cell are $h_{top}$ and $h_{down}$, and the corresponding container marking distance for storing the cell is d. At this time, radial elevation expansion is performed on the cell; the maximum value of the corresponding elevation of the cell in the container is $h_{max} = h_{top} + \sqrt{R^2 - (r^2 * d)/3}$ and the minimum value is $h_{min} = h_{down} - \sqrt{R^2 - (r^2 * d)/3}$. Through decoding, the maximum and minimum values of the SGOG radial ID are obtained, and all grid cells of SGOG located in the buffer are obtained according to the continuity of the SGOG radial ID.

Figure 4 is a schematic diagram of the radial elevation expansion, assuming that there is only one initial cell with ID 0000 and edge length $6\sqrt{3}$. The upper elevation of the initial cell is 1 and the lower elevation is -1, so the initial cell could be expressed as $(0000, 1, -1)$; its similar triangular prism structure is shown in Figure 4a. By carrying out edge neighbor expansion according to the above method, the cell IDs obtained after one edge neighbor expansion are 0001, 0002, and 0003, which are expressed as $(0001, 1, -1)$, $(0002, 1, -1)$, and $(0003, 1, -1)$, and the structure is shown in Figure 4b. Then, the Euclidean distance from the cells with IDs 0001, 0002, and 0003 to the initial cell ID 0000 can be calculated to be 6, and if the buffer radius is 10, the elevation expansion value of the corresponding cells is calculated to be 8. Finally, the elevation is expanded up and down as the calculations,

which is shown in Figure 4c. The cells in the 3D buffer are expressed as (0000, 11, −11), (0001, 9, −9), (0002, 9, −9), and (0003, 9, −9), as shown in Figure 4d.

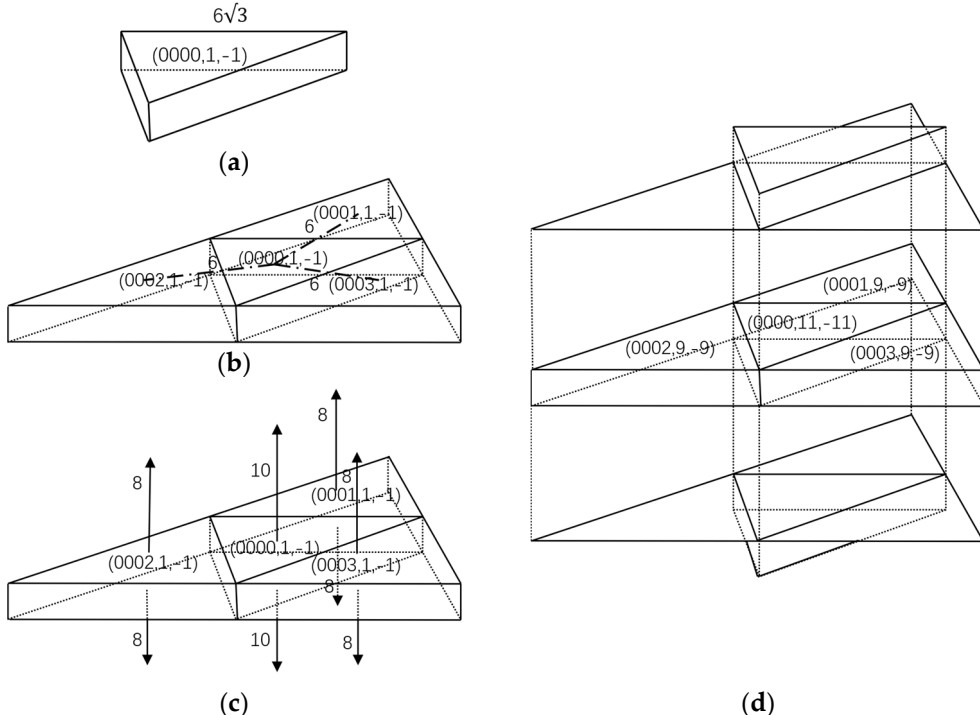

**Figure 4.** The radial elevation expansion of a cell. (**a**) The structure and the expansion direction of an initial cell; (**b**) The structure of the cells obtained by one edge neighbor expansion; (**c**) The radial elevation expansion direction of the cells; (**d**) The structure of the cells obtained by radial elevation expansion.

*2.3. The Generation of the 3D Buffer*

The purpose of 3D buffer generation of a point is first to transform the target point to the SGOG; that is, the initial cell is a grid cell of SGOG. The corresponding data are a cell ID and its corresponding elevation, and the Euclidean distance expansion based on QTM edge neighbors is performed on the cell ID to obtain cells and the Euclidean distances from these cells to the initial cell. The spherical 2D buffer of the cell ID is obtained as shown in Figure 5a below. Then, through the radial elevation transfer expansion, that is, the elevation of the cell ID of the spherical 2D buffer zone corresponding to the radial expansion of the buffer radius, the 3D buffer zone of the cell is obtained as shown in Figure 5b below.

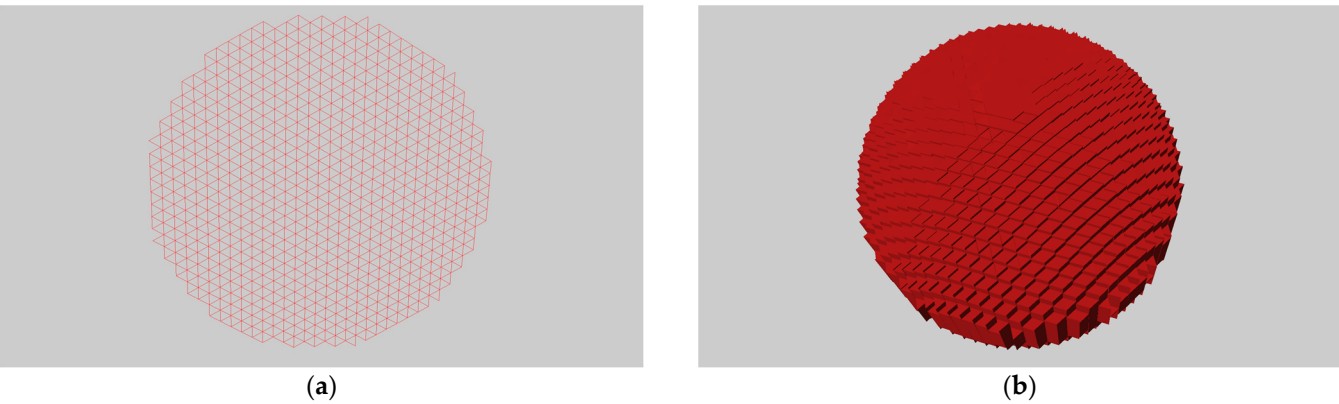

**Figure 5.** (**a**) The spherical 2D buffer of a point; (**b**) The sphere 3D buffer of a point.

To generate the 3D buffer of a line object, the line object needs to be transformed to the SGOG first. In this paper, the dimensionality reduction generation algorithm of the triangular mesh discrete line by Zhao et al. [34] was used to discretize the spherical triangular mesh of a vector line. Then, the cells of the entire line in the SGOG were expanded in a 3D buffer, and the cell IDs and upper and lower elevation data of the buffer were obtained. Different initial cells may produce cells with repeated cell IDs when the 3D buffer is expanded, and because the corresponding initial cells of the repeated cells are different, the upper and lower elevation data of the repeated cells will be different. Therefore, it is necessary to obtain the union of the obtained buffer cells and perform data fusion on the repeated cells. For the cells with repeated cell IDs in all the obtained buffer cells, only one cell corresponding to the cell ID was retained. The upper elevation of this retained cell ID is the maximum value of the elevation subset on the repeated cell, and the lower elevation of it is the minimum value of the elevation subset on the repeated cell. In summary, 3D buffer generation of the line was completed, and the effect diagram is shown in Figure 6.

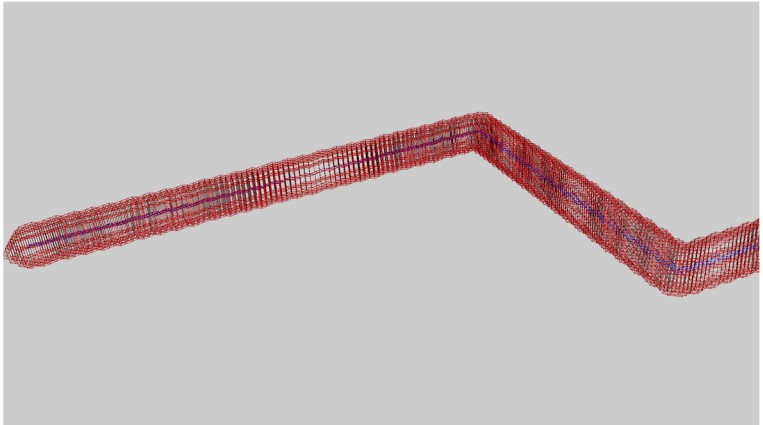

**Figure 6.** The 3D buffer of a line. Cells marked as the target line are shown in blue, and cells marked as the buffer are shown in red.

With regard to the 3D buffer generation algorithm for area objects and volume objects, in view of the similarity between an area object and a volume object when they are transformed to the SGOG, the area object is a volume object with a very small difference between the upper and lower elevations, and the generated 3D buffers are also volume objects. Therefore, the 3D buffer generation algorithms for the two are the same. First, the boundary of the target DEM data is extracted to obtain its boundary line. The 3D buffer generation algorithm of a line object is adopted for the boundary line to obtain the peripheral buffer area of the boundary line. The peripheral buffer area refers to the buffer area obtained by the boundary line expanding outward to the target DEM. The target DEM is transformed to the cell IDs of the SGOG, and the corresponding DEM data of the obtained cell IDs are matched to the corresponding elevations. Therefore, to complete the target object transformation to the SGOG, radial elevation expansion is performed on the corresponding buffer radius of the acquired grid cells of the SGOG; that is, the upper and lower elevation values of the corresponding cell IDs correspond to the addition and subtraction of the buffer radius, respectively, to obtain the target's radial elevation buffer area. Taking the union with the peripheral buffer area of the boundary line, the result is the 3D buffer of the target. In summary, taking the generation of the 3D buffer of the volume object as an example, the overall flow of the algorithm is shown in Figure 7.

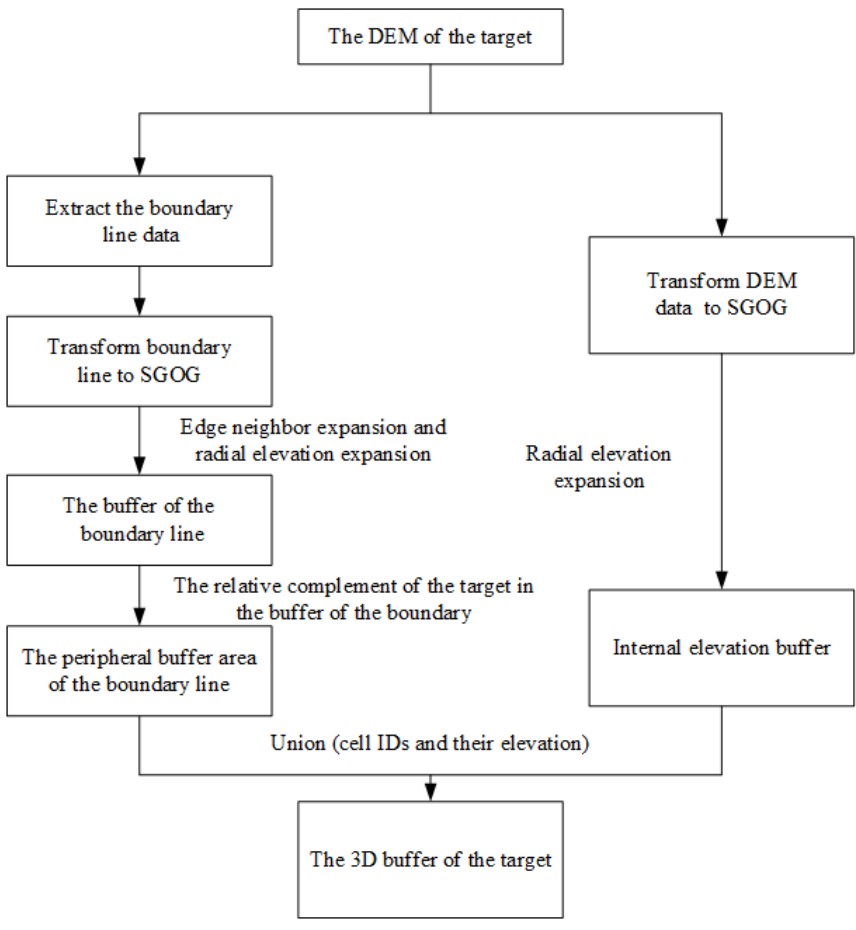

**Figure 7.** Flow chart of volume object 3D buffer generation.

## 3. Results

In this section, we present a visualization (Section 3.1) and the efficiency and complexity of the algorithm (Section 3.2) by applying the abovementioned methods to generate a 3D buffer.

### 3.1. Visual Analysis of the Experiment

The DEM data of a certain area in Xinyang City, Henan Province were selected to test the feasibility of the 3D buffer generation algorithm for volume objects. The experimental equipment used was a DESKTOP-62NRMKN desktop computer; hardware configuration: Intel (R) Core (TM) i7-7700H CPU @ 3.60GHz, RAM 32.00GB; operating system: Win10 Professional Edition; development language: C++; development tool: Visual Studio 2017; 3D Engine API: OpenSceneGraph (OSG).

First, it was determined that the sphere grid level of this experiment was 18 layers, the corresponding grid triangle side length in the SGOG was approximately 40 m, the radial grid level was 22 layers, and the corresponding grid radial length in the SGOG was approximately 1.5 m. A 3D visualization of its grid in SGOG is shown in Figure 8a below. The radii of the buffer zone selected in the experiment were 40, 120, and 200 m. The 3D visualization effect wireframe of the buffer zone is shown in Figure 8 below, and cross-sectional views are shown in Figure 9. The blue color in the figure represents the target space object, and the red color represents the buffers with the corresponding radii.

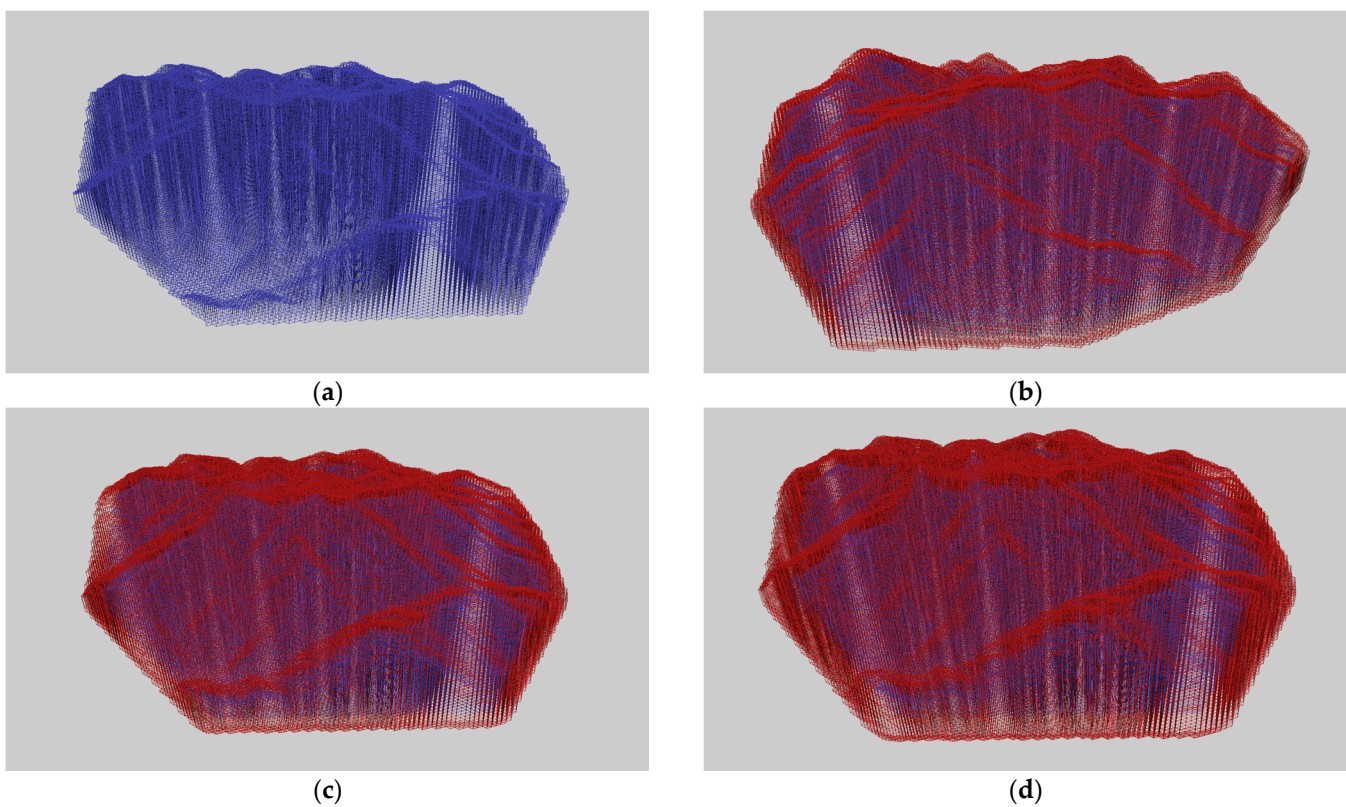

**Figure 8.** Wireframe models of the target area (**a**) and its buffers with 40 (**b**), 120 (**c**), and 200 m (**d**) radii at the sphere grid level of 18. Cells marked as the target area are shown in blue, and cells marked as the buffer are shown in red.

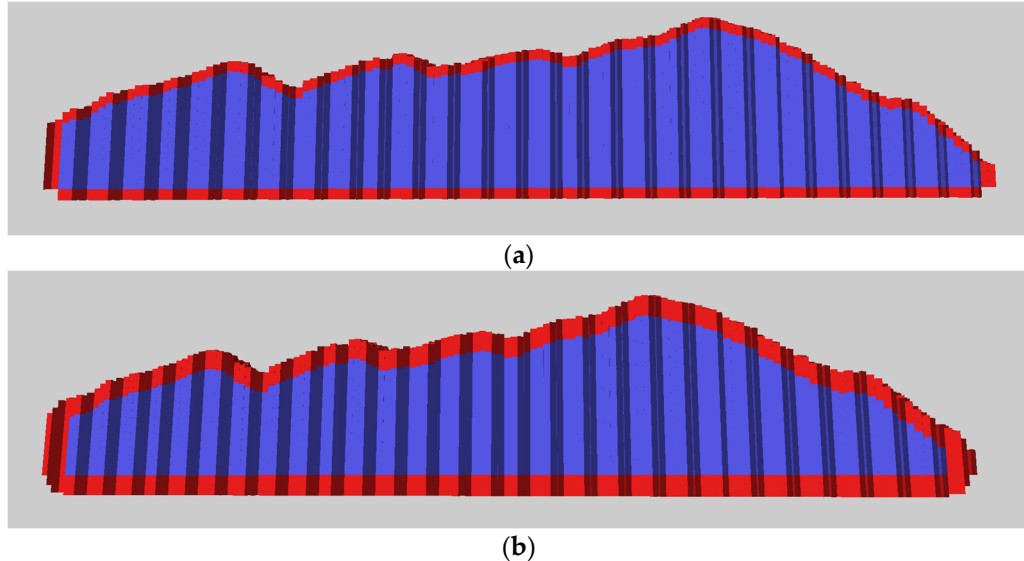

**Figure 9.** *Cont.*

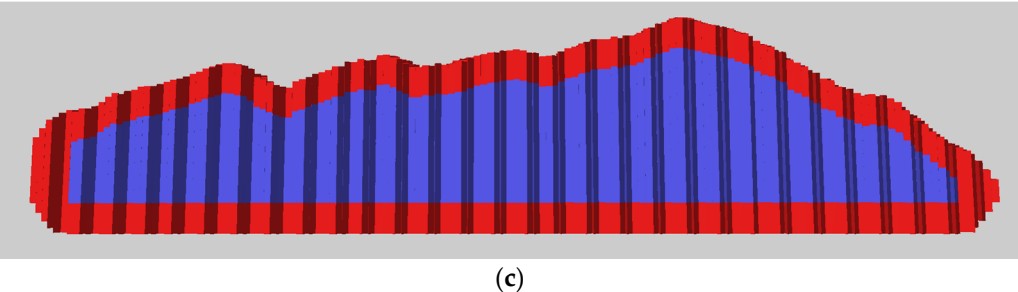

(**c**)

**Figure 9.** Cross-sectional views of the target area and its buffers with 40 (**a**), 120 (**b**), and 200 m (**c**) radii at the sphere grid level of 18. Cells marked as the target area are shown in blue, and cells marked as the buffer are shown in red.

Sphere grid levels of 18 (triangular grid side length of approximately 40 m) and 19 (triangular grid side length of approximately 20 m) layers were selected to compare the visualization effects of the 3D buffer with a buffer radius of 80 m. The wireframe and cross-sectional views are shown in Figures 10 and 11 below. When the sphere grid level was 18, the number of cells for the test area was 23,372, and the number of cells in the 3D buffer obtained by the radial elevation expansion was 23,372 × 2. The number of cells for the boundary line was 838, and the number of cells which were transformed by the peripheral buffer area of the boundary line obtained by edge neighbor expansion and radial elevation expansion was 2526. In summary, the number of cells in the 3D buffer was 49,270. However, when the sphere grid level was 19, the number of cells in the test area was 94,317, and the number of cells in the 3D buffer obtained by radial elevation expansion was 94,317 × 2. The number of cells in the boundary line was 1688, and the number of cells which were transformed by the peripheral buffer area of the boundary line obtained by edge neighbor expansion and radial elevation expansion was 8420. In summary, the number of cells in the 3D buffer was 197,054. It can be seen that the greater the sphere grid level, the smaller the side length of the triangle grid, the greater the number of cells in the grid, and the higher the accuracy of the buffer.

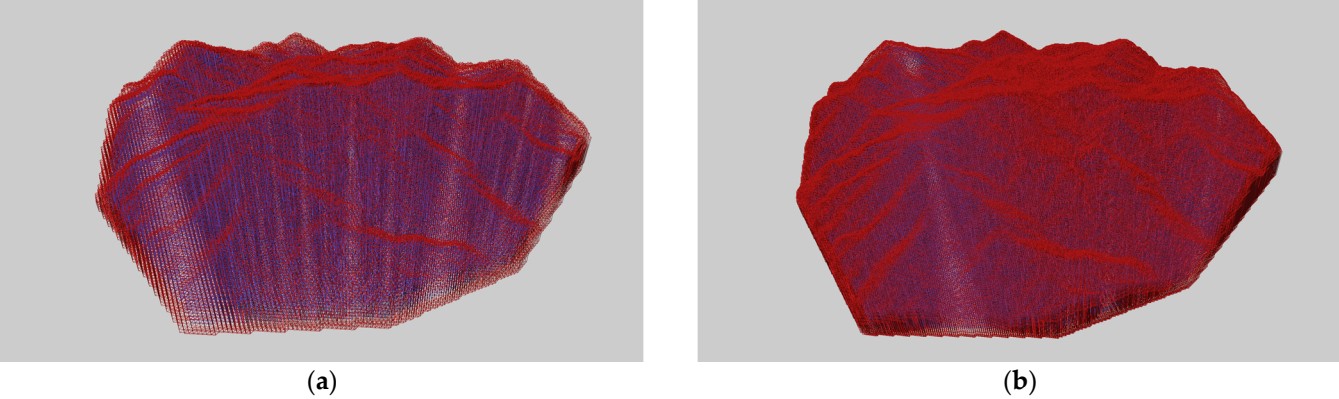

(**a**)                    (**b**)

**Figure 10.** Wireframe models of the target area and its buffer with 80 m radius at the sphere grid levels of 18 (**a**) and 19 (**b**). Cells marked as the target area are shown in blue, and cells marked as the buffer are shown in red.

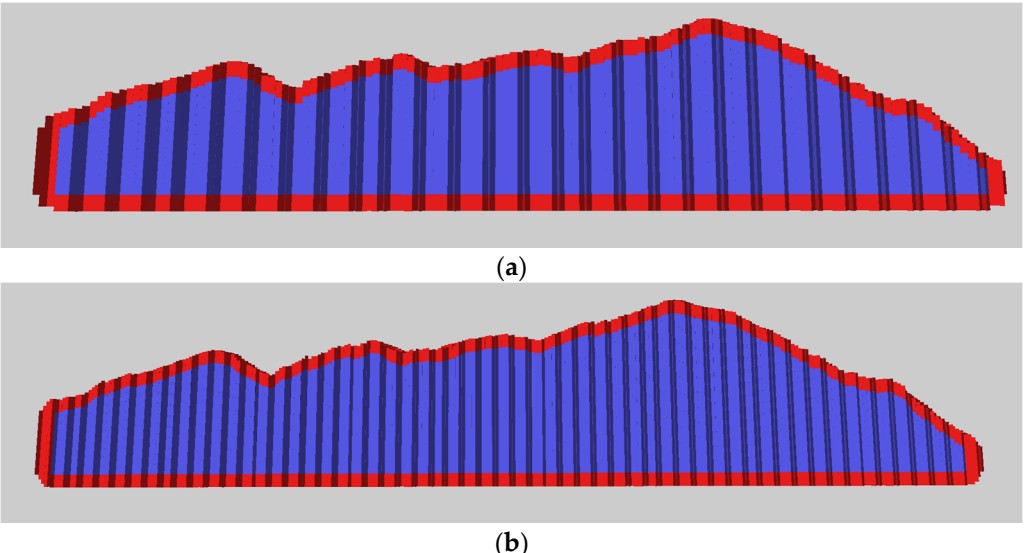

**Figure 11.** Cross-sectional views of the target area and its buffer with 80 m radius at the sphere grid levels of 18 (**a**) and 19 (**b**). Cells marked as the target area are shown in blue, and cells marked as the buffer are shown in red.

### 3.2. Algorithm Complexity Analysis

The time complexity of the point 3D buffer generation algorithm is composed of the time complexity of the Euclidean distance expansion based on the QTM edge neighbors of the initial cell and the time complexity of the radial elevation expansion algorithm. For a 3D buffer with radius R, the number of cells in the grid obtained by the edge neighbor expansion algorithm is N, where N is proportional to $R^2$, so the time complexity of the edge neighbor algorithm is O(N), which is O($R^2$). The radial elevation transfer expansion algorithm is the same, and its time complexity is O(N), which is O($R^2$). Therefore, the time complexity of the 3D buffer generation algorithm is O($R^2$). Figure 12 shows the algorithm efficiency diagram for the 3D buffer of a point. In this experiment, the sphere grid level of the SGOG was 23 layers (triangular grid side length was approximately 1 m). The abscissa is the radius of the buffer, and the ordinate is the calculation time of the 3D buffer. It can be seen from Figure 12 that the generation time of the 3D buffer is roughly proportional to the square of the buffer radius, which verifies that the time complexity of the 3D buffer generation algorithm for the point is O($R^2$). The time complexity of the traditional grid-based isosurface expansion method to generate a 3D buffer is O($R^3$) [35], so the efficiency of the proposed method is significantly improved.

The time complexity of the 3D buffer generation algorithm for volume objects is shown in Figure 13. This figure was obtained from the relationship between the radius and time for the 3D buffer generated for the geological body in Section 3.1 using different sphere grid levels. For a sphere grid level of 18 (triangular mesh side length of approximately 40 m), the buffer generation times with radii of 40–400 m were calculated. For a sphere grid level of 19 (triangular mesh side length of approximately 20 m), the buffer generation times with radii of 20–100 m were calculated. The specific generation times are shown in Tables 3 and 4. The number of spherical triangle grids for a sphere grid level of 19 was approximately four times that for a sphere grid level of 18. Although the accuracy of the buffer was improved, it can be seen from Figure 13 that the multiplication of the spherical triangular grid greatly increased the generation time of the buffer. The time complexity of the volume buffer is due to the internal radial elevation buffer and the boundary peripheral buffer. At the same sphere grid level, the number of internal regions and boundary grids is determined, so the time complexity of the internal radial elevation buffer is constant, independent of the radius. The time complexity of the boundary peripheral buffer is O($R^2$),

so its time complexity is $O(R^2)$, which is verified by the relationship between the radius of the buffer with a sphere grid level of 18 and the time consumed in Figure 13.

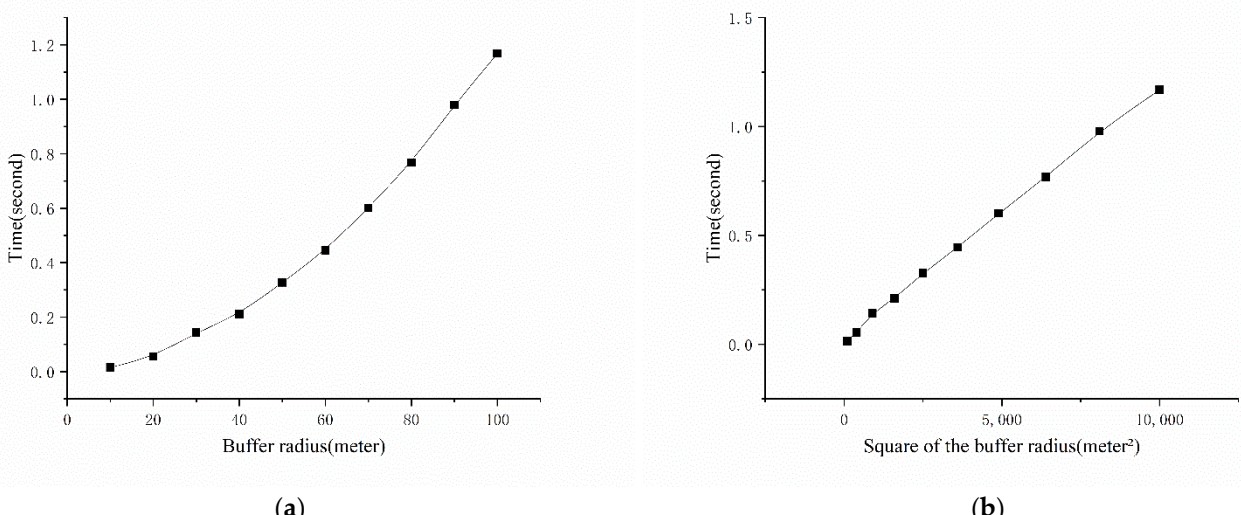

(**a**) (**b**)

**Figure 12.** Algorithm efficiency diagrams for a 3D buffer of a point: (**a**) The X-axis represents the buffer radius, and the Y-axis represents the computation time. (**b**) The X-axis represents the square of the buffer radius, and the Y-axis represents the computation time.

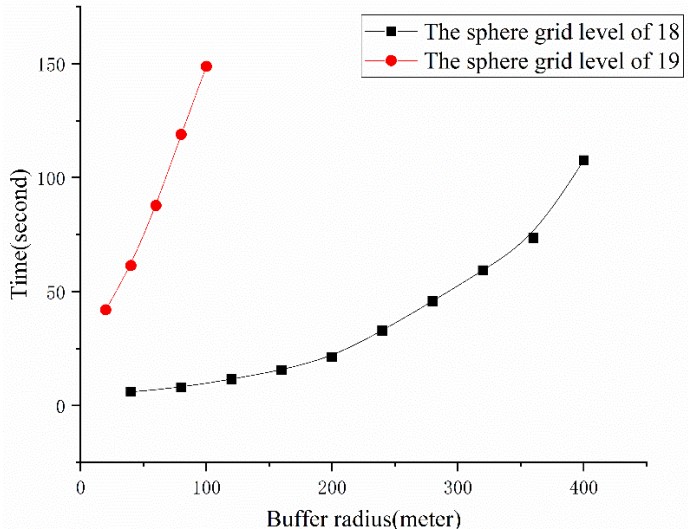

**Figure 13.** Comparison of the algorithm efficiency of the target area for sphere grid levels of 18 and 19. The X-axis represents the buffer radius, and the Y-axis represents the computation time.

**Table 3.** The radii and generation times of the buffers with a sphere grid level of 18.

| Buffer Radius (m) | 40 | 80 | 120 | 160 | 200 | 240 | 280 | 320 | 360 | 400 |
|---|---|---|---|---|---|---|---|---|---|---|
| Generation Time (s) | 5.982 | 8.019 | 11.57 | 15.56 | 21.31 | 32.86 | 45.81 | 59.31 | 73.48 | 107.59 |

**Table 4.** The radii and generation times of the buffers with a sphere grid level of 19.

| Buffer Radius (m) | 20 | 40 | 60 | 80 | 100 |
|---|---|---|---|---|---|
| Generation Time (s) | 41.993 | 61.401 | 87.79 | 118.983 | 148.794 |

## 4. Conclusions

The development of 3D DGGS provides new opportunities for 3D space analysis. In this paper, we proposed a method for generating 3D buffers based on the 3D DGGS, taking into account the characteristics of raster algorithms and 3D DGGS. Through time complexity analysis, the time complexity of the algorithm in this paper was determined to be $O(R^2)$, while the time complexity of the raster algorithm is $O(R^3)$ [35]. The concept of dimensionality reduction in the calculations was adopted, and the spherical topological characteristics and radial continuity of the SGOG were used. The calculation of the 3D buffer was converted to the generation of a 2D spherical discrete grid buffer and the calculation of the corresponding radial elevation. The time complexity of the algorithm was reduced, thereby improving the efficiency of the algorithm. With increasing sphere grid level of the SGOG, the number of cells in the grid increased geometrically. In practical applications, a balance should be struck between analysis accuracy and algorithm efficiency. While still meeting the accuracy requirements, a smaller sphere grid level should be chosen to effectively improve algorithm efficiency. It is important to mention that the 3D buffer generation methods presented in this paper are suitable for diamond 3D DGGS and hexagonal 3D DGGS, for which neighbor expansion based on complex units and dimensionality reduction would be similar to the methods for the SGOG.

There are still some important future works to be done. In terms of modeling the 3D buffer, our next step is to explore multiresolution modeling approaches for triangular 3D DGGS, whereby the object is represented as a multiresolution set of cells instead of a single-resolution set of cells, which may reduce the number of the cells in the grid. Subsequently, we intend to explore multiresolution generation and analysis approaches for 3D buffers based on multiresolution modeling of the object to reduce the number of 3D DGGS cells, thereby further improving the efficiency of the algorithm. Since 3D DGGS cells are ideally suited for integrating big geospatial data at different spatial resolutions, we anticipate that such an approach could prove to be feasible.

**Author Contributions:** Conceptualization, Jinxin Wang and Yan Shi; methodology, Zilong Qin and Yihang Chen; software, Zening Cao; validation, Jinxin Wang, Yan Shi and Yihang Chen; formal analysis, Zilong Qin; investigation, Zening Cao; resources, Jinxin Wang; data curation, Yan Shi; writing—original draft preparation, Yan Shi; writing—review and editing, Yan Shi; visualization, Yan Shi and Zening Cao; supervision, Jinxin Wang and Yihang Chen; project administration, Jinxin Wang.; funding acquisition, Jinxin Wang. All authors have read and agreed to the published version of the manuscript.

**Funding:** This research was funded by Key Scientific and Technological Project of Henan Province, China, grant number 212102210377.

**Institutional Review Board Statement:** Not applicable.

**Informed Consent Statement:** Not applicable.

**Data Availability Statement:** Publicly available datasets were analyzed in this study. The DEM data was provided by Geospatial Data Cloud site, Computer Network Information Center, Chinese Academy of Sciences: http://www.gscloud.cn (accessed on 2 June 2021).

**Acknowledgments:** We are grateful for the comments of the anonymous reviewers, which greatly improved the quality of this paper.

**Conflicts of Interest:** The authors declare no conflict of interest.

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
