# Peer review of "A Three-Dimensional Buffer Analysis Method Based on the 3D Discrete Global Grid System"

_ijgi, doi:10.3390/ijgi10080520_

Round 1
Reviewer 1 Report
Line 99: The last paragraph of the Introduction needs to be improved. Why is the 3D geographic computing important? What is the goal of this study? What are the challenges that you are trying to address? In addition, is there any reason that you selected SGOG rather than the others?
Line 105: What is the scientific significance of this study? Why is this study important?
Line 107: You should add a new paragraph to briefly introduce each section.
Line 108: What you proposed and what previous studies had done need to be clear. What are the parts that you newly developed or suggested? What are the methods or formulas that you used in previous studies?
In addition, the Methods section looks more like a report. You should provide justification or reasons for which you suggested in that way or used a specific method (e.g., why did you separate the edge proximity into the positive and negative? What is the role of DEM in making 3D buffers? What are the benefits of using QTM side neighbors? and etc.).
Line 304: Is the DEM ‘date’ right in Figure 7?
Line 335: You need to quantitatively assess the accuracy of the buffer and show the result. The judgement for the accuracy based on its visualization is not sufficient.
Line 382 and 383: /m -> (m), /s -> (s)
Line 384: It seems you received feedback from somewhere and put the comments without resolving the issues. I agree with that comments. You should discuss the findings and provide interpretation. Implications of this study and future work need to be added.
Author Response
We gratefully thank you for the time spend making your constructive remarks and useful suggestions, which has significantly raised the quality of the manuscript and has enable us to improve the manuscript. Each suggested revision and comment are response point by point and the revisions are indicated. Please see the attachment for detailed responses.

Reviewer 2 Report
1) MDPI Style Guide: "Journal references must cite the full title of the paper, page range or article number, and digital object identifier (DOI) where available." Many of the citations are incomplete. Please check ALL refs for completeness, these are a few examples: No references have DOI; e.g. Ref 1 is missing the issue number: 44 should be 44(9); Refs 17, 18 are missing issue number and page numbers.
2) [p2 lines 74-96] and elsewhere. As I understand it the acronym ESSG was originally introduced (~2011) to distinguish 3D approaches from 2D DGGS - which was a term that had been in use for a decade or two. The useage of DGGS has since expanded to include 3D, and 4D spatio-temporal reference systems [cf OGC Topic 21 v2.0/ISO 19170-1:2021 DGGS Core https://www.iso.org/standard/32588.html or Weixin Zhai et al 'Collision Detection for UAVs Based on GeoSOT-3D Grids' https://www.mdpi.com/2220-9964/8/7/299 ]. Even by 2013 [ref 27] the distinction was reducing, so I suggest that the use of the term ESSG as something that is different from DGGS [Line 85-86], may have been true in 2012 [ref 25] but is now missleading and confusing. So, I recommend saying that ESSG and DGGS are now essentially the same thing.
3) [p2 line 64 etc] & [p3 line 104 etc] useage of the term 'grid' vs 'cell' and usage of the terms 'sphere code'|'grid code' vs 'cell ID'. Use of the term 'cell' helps to distinguish between a whole grid or just a single cell within the grid. Also 'cell identifier', or 'cell ID' for short, or sometimes or 'cell index' are all more widely used than 'grid code' or 'sphere code'. So a grid is constructed from many cells, and each cell has a cell ID. So I would suggest changing 'sphere code' and 'grid's code' to 'cell ID' throughout, and where a single cell is referred to change 'grid' to 'cell', and where a small number of cells are referred to change 'grids' to 'cells'. For example [p2 line 64] change 'greater the number of grids' to 'greater the number of cells in the grid', [p3 line 104] change 'sphere codes' to 'cell IDs', and [p3 118-121] change sentence to: 'Then, the transformed SGOG cell IDs were transformed by the Euclidean distance expansion algorithm based on QTM cell edge neighbors, and the corresponding cell IDs and the Euclidean distance relationship of the corresponding target cells were obtained, namely, the 2D spherical buffer of the target object.' etc throughout.
4) [p3 line 102 etc] change 'side neighbor(s)' to 'edge neighbor(s)' throughout. The sides of cells are referred to as 'edges', especially when topological relationships such as neighbor are used.
5) [p3 line 114] Change 'First, the target Digital Elevation Model (DEM) data ..' to 'A Digital Elevation Model (DEM) was used as the test target for our 3D buffer operations. First, the target DEM data ..'. This is the first reference to DEM data as the buffer target, so a little more clarification of what the DEM data is to be used for is very helpful to the reader.
6) [p3 line 114-126] This is the first time the proposed buffer technique is described. I suggest a diagram, or maybe two simple diagrams illustrating a) the expansion in 2D, and b) the radial elevation in 3D, would really help readers. The diagram should show the buffer on a very simple 3D target represented by a minimal number of SGOG cells, possibly three neighboring cells, with two cells at the same radial elevation, and one cell above (or below) the other two. You could even refer to the cell IDs from the diagrams in the text at each step. This would help the reader enormously to get a clear picture in their mind right at the start of the method. Fig 4 could be brought forward, especially if it was modified as suggested below.
7) [p3-9 Sect 2] Section 2's sub-section numbering has got corrupted:
[p3 line 127] change '3.1' to '2.1',
[p3 line 143] change '3.2' to '2.2',
[p4 line 153] change '3.1.1' to '2.2.1',
[p7 line 231] change '3.1.2' to '2.2.2',
[p8 line 256] change '3.3' to '2.3'.
8) [p5 line 180] change 'Equation 1' to Equation (1)' to be consistent.
9) [p5 line 182-3, and 189] insert Oxford comma ',' before each last pair of items as follows:
182-3: 'v11 and v23, v12 and v21, and v13 and v22'.
189: 'v11 or v23, v12 or v21, and v13 or v22'.
10) [p6 line 208-219, and Fig 3.] The text in this paragraph doesn't match the sequence of red arrows in Fig 3. The sequence of red arrows shows two possible paths from 102->113, and paths from 103->131, and 110->111, but the text [line 216-7] only mentions 102->112,110,113, and the text does not mention the paths fom 103->131 or 110->111. If these unmentioned red arrows are unnecessary for the example in the text, then delete them from the Fig 3 or if they are necessary, then explain why. Also the sentence in [line 216-7] would be better phrased 'In the figure, the path from cell ID 120 to cell ID 113 is the complex unit sequence v21, v11, v21, v11 and v22.
11) [p6 lines 219] The maths 'l=√13/3r' would be clearer as 'l=r√13/3', so that 'r' is in the same place as it is in Equation (6).
12) [p7 lines 232-253, and p8 figure 4] In the figure change 'pattern 1', 'pattern 2' ..'pattern 4' to (a), (b) .. (d), so it is the same as other diagrams and add short descriptions of each of (a)..(d) to the Fig 4 caption, and then change the references in the text from '4.1, '4.2' etc to '4.a', '4.b' etc. so that they are referenced in the same way as other figures such as fig 5.a.
[Figure 4, pattern 3] Show 6 arrows pointing up or down. It would help if these could have different lengths representing the values of hmax and hmin for their cells. Similarly if [Figure 4, pattern 2] could have an arrow of length 'r' that would further demonstrate the difference between QTM and radial expansion. [Figure 4, pattern 4] would be helped by the addition of cell IDs for the two cells that are added as by the radial expansion - one above and one below cell 0000. Then at [p7 line 252-3] these new cell IDs could be mentioned in the text.
13) [p8, figure 5] The cell size used for Figure 5.a and 5.b is too small to be clearly seen at the size and level of detail provided for review. While this level of detail produces a pleasantly smooth circular and spherical edge, there are too many cells and the lines too fine for the image resolution in the pdf file. The same applies to Figures 6, 8, 9, 10 and 11. Please discuss how to resolve this with the MDPI.
14) [p11 Figure 8, p12 Figure 11] These figures show vertical dark (black) bands at regular intervals that are a distraction. This may be an artifact related to my (13) above, or it may be due to some cells having vertical faces that coincide with the plane of the cross-section - or a combination of both. If this is the case.
15) [p12 line 330-339, Figure 10, Figure 11] While it is certainly useful to test running the buffer algortithm at different reolutions (18 vs 19) as in [Section 3.2], I think that it isnt necessary to show the results in figures just to demonstrate the conclusion 'It can be seen that with the increase in the sphere grid level, the smaller the side length of the triangle grid, the greater the number of grids, and the higher the accuracy of the buffer.' [Lines 333-5]. I would only show the comparison between two levels if some other useful and maybe unexpected information could be shown.
16) [p13 line 349] change 'gird' to 'grid'.
17) [p14 lines 384-386] Please remove this author suggestion, and check whether there is some missing text that you intended to place here.
18) [p13, 14] I think a major opportunity has been lost, by not discussing potential algorithms that use the multi-resolution nature of DGGS (and ESSG) to gain execution efficiency. You refer to Bowater [24], who does this for the 2D DGGS rHEALPix. 2D SGOG has the same nesting characteristics as rHEALPix, so 2D SGOG would benefit from the a very similar approach and I think it would be very worthwhile to think about applying multi-resolution approaches to solving the radial extension as well for 3D SGOG. I'm not suggesting that you do more work before publishing, but I think [Figure 13] demonstrates the need for further improvement in efficiency, and so you should at least discuss this in the Conclusion.
Author Response

(The authors gave the same response as above.)

Reviewer 3 Report
This paper presents a new method for the creation of a 3D buffer on ESSG. The research is interesting but a clearer presentation of the methods is needed to support it.
line 155: is the euclidean distance expansion method based on existing theory or methods?
line 159: what is the necessity of the complex unit?
line 164: what are positive and negative grids?
Equation 1: which angle is equal to π/6?
line 178: what are steps of complex units? what is the use of ko - k5?
line 182 - 191: please explain the deletion procedure
Equation 4: what are x,y,z? the cell coordinates?
Figure 2: explain colors used
Line 205: what is the initial grid distance l?
Line 235: what is the grid and what the initial grid?
Line 235-244: text is rather comfusing!!!
change Figure 4 subtitles according to lines 245 -253
Figure 6 does not seem 3D.
Figure 7. what is the boundary line date? Enhance this graph to provide a summarizing image of the methods
line 316: what are layers?
Figure 8. images should be larger. Maybe they need to be redesigned to portray better the 3D/
Good luck in publishing the paper.
Author Response

(The authors gave the same response as above.)

Reviewer 4 Report
The authors introduced a novel method of creating a 3D buffer based on ESSG,
In their paper, SGOG was selected as a data storage. The 3D buffer generation was based on Quaternary triangular mesh side neighbours.
The authors in the methods section meticulously describe the workflow applied in the generation of 3D buffer.
In the results section is visual analyses were performed, the figure adequately supports the results, and an algorithm complexity analysis was carried out.
What is for me completely missing the kind of discussion part where authors with exact comparison can prove that their method is really more efficient than any other method, it can be implemented into both result part, visually also they can prove that their method better, and also they can make a comparison that the computation capacity and the time use is more efficient using their method they can add this information into table 3 and 4 as well.
The discussion section is completely missing and also a conclusion is too short
Minor issues
- line 274-276 why union was used instead of dissolve method.
- line 384-387 -it seems these lines are coming from previous review results (I completely agree that this part is still missing from the paper)
- figure 6 maybe buffer can be transparent and we can see it, I this form it is meaningless, we can see only a thick line
Author Response

(The authors gave the same response as above.)

Round 2
Reviewer 1 Report
Now, it looks better than before to be published.
Reviewer 3 Report
Authors have revised the paper as suggested.
Reviewer 4 Report
Dear Authors, thank you for the clarification.
The visualisation is much better, although, still it is not clear why this method is better.